# Evaluating the Harms of Cancer Testing—A Systematic Review of the Adverse Psychological Correlates of Testing for Cancer and the Effectiveness of Interventions to Mitigate These

**DOI:** 10.3390/cancers15133335

**Published:** 2023-06-25

**Authors:** Fong Lien Kwong, Clare Davenport, Sudha Sundar

**Affiliations:** 1Institute of Applied Health Research, University of Birmingham, Edgbaston, Birmingham B15 2TT, UK; c.f.davenport@bham.ac.uk; 2Institute of Cancer and Genomic Sciences, University of Birmingham, Edgbaston, Birmingham B15 2TT, UK; s.s.sundar@bham.ac.uk

**Keywords:** anxiety, cancer, diagnosis, harm, interventions, psychological

## Abstract

**Simple Summary:**

There is a drive to detect cancers at an early stage to improve survival. While this initiative has been associated with better outcomes for certain cancers, testing also leads to patient anxiety and distress. Most of the research in this domain was conducted in asymptomatic patients who attend as part of population-based testing (screening). The literature in individuals with symptoms or with abnormal preliminary results (diagnostics) remains deficient. We conducted a literature search to identify which cancers were underrepresented, what risk factors could contribute to worse psychological outcomes in both screening and diagnostics, and whether any interventions could help to mitigate these. Our search revealed that young, unemployed individuals were at high risk and should therefore be targeted for support. Among the interventions considered, the use of patient leaflets, one-stop clinics, and patient navigators to facilitate patient attendance at their appointments appeared to be the most beneficial.

**Abstract:**

(1) Background: Several studies have described the psychological harms of testing for cancer. However, most were conducted in asymptomatic subjects and in cancers with a well-established screening programme. We sought to establish cancers in which the literature is deficient, and identify variables associated with psychological morbidity and interventions to mitigate their effect. (2) Methods: Electronic bibliographic databases were searched up to December 2020. We included quantitative studies reporting on variables associated with psychological morbidity associated with cancer testing and primary studies describing interventions to mitigate these. (3) Results: Twenty-six studies described individual, testing-related, and organisational variables. Thirteen randomised controlled trials on interventions were included, and these were categorised into five groups, namely the use of information aids, music therapy, the use of real-time videos, patient navigators and one-stop clinics, and pharmacological or homeopathic therapies. (4) Conclusions: The contribution of some factors to anxiety in cancer testing and their specificity of effect remains inconclusive and warrants further research in homogenous populations and testing contexts. Targeting young, unemployed patients with low levels of educational attainment may offer a means to mitigate anxiety. A limited body of research suggests that one-stop clinics and patient navigators may be beneficial in patients attending for diagnostic cancer testing.

## 1. Introduction

Medical tests to detect cancer are key to improving early diagnosis and improving oncological outcomes, including patient survival. The Faster Diagnostic Framework was set up by NHS England in the U.K. in 2015 to fast-track patients with a possible diagnosis of cancer [1]. One of the aims of this initiative is to reduce anxiety associated with prolonged waiting times, especially for patients irrespective of their diagnosis. Although testing is commonly viewed as beneficial, testing can cause harm. Harms associated with testing may be direct (for example pain associated with the test application and anxiety) or indirect, for example the harms associated with the downstream consequences of a test result including test errors (false positives and false negatives).

The context of testing (screening or diagnosis) and the place of a test in the clinical pathway (early or late) will determine the nature and importance of downstream indirect consequences. Screening usually refers to routine testing in asymptomatic average risk individuals to evaluate their risk of developing cancer. Diagnostic tests, in contrast, principally refer to testing to determine whether at-risk individuals actually have cancer. Whilst the purpose of screening and diagnostic tests are different, it follows that a screening test may lead to diagnostic testing in individuals who are identified as being at increased risk of developing cancer. For example, a missed cancer diagnosis (false negative, FN) may be afforded greater importance than a false positive (FP) result for an individual undergoing diagnostic testing. However, when tests are applied in low-prevalence populations such as in screening, the consequences of a missed cancer diagnosis (FN) need to be balanced against the consequences of receiving an FP for a larger absolute number of individuals. Research to date has largely concentrated on the therapeutic, financial, psychosocial, and legal implications that occur as a result of cancer-screening programmes [2]. In contrast, the consequences associated with diagnostic testing for cancer have received less attention.

There is compelling evidence that a negative testing experience per se may have a detrimental impact on patient satisfaction and reduce motivation to engage with healthcare services or attend for further testing or treatment. Studies have demonstrated a potential link between the level of psychological distress and the strength of the body’s immune system [3,4].

With various initiatives that will result in an increase in the number of individuals undergoing diagnostic testing for cancer, it is important to understand the potential psychological impacts of testing policy. In addition, determining whether certain individuals are more vulnerable to the adverse psychological effects of testing would allow targeting of interventions to mitigate these.

### Existing Research

We sought to identify any systematic review concerned with quantifying the psychological associations of cancer testing and the effectiveness of interventions to mitigate this. A scoping search conducted in December 2020 across systematic reviews evaluating the psychological associations of cancer testing across Ovid MEDLINE and Embase yielded a single quantitative review [5] which examined the levels of anxiety, stress, worry, panic, and fear associated with screening tests for breast, colon, prostate, and lung cancers pre-test, post-test, and post-negative-test results. Only studies conducted in the United States and published between 1946 and October 2016 were included. The authors excluded studies about cancer testing in a diagnostic context and confined their review to examination of the consequences of positive test results.

We therefore undertook a review with the aim of addressing deficiencies in the literature apropos of an up-to-date review without geographical restriction considering the psychological associations of testing for cancer and the potential effects of the entire testing process (pre-, during, and post-) regardless of test result. We also sought to ascertain evidence about interventions that mitigate anxiety in individuals undergoing cancer testing. Through this review, we also aim to highlight which cancers have been the most well-researched to date and thereby identify the types of cancer where a paucity of evidence prevails and where further research is mandated.

We anticipated a paucity of the literature concerned with diagnostic testing as opposed to screening and therefore decided to include both types of test application in our review scope. Whilst we hypothesised that there may be overlap in mechanisms of psychological associations and effectiveness between screening and diagnosis, we acknowledged potential differences by test application by distinguishing these in our synthesis.

## 2. Materials and Methods

### 2.1. Review Questions and Inclusion Criteria

Two separate frameworks for question formulation were used: SPIDER [6] for question 1, as this was concerned with a phenomenon that could be evaluated using diverse research approaches, and PICO for question 2, which is concerned with the examination of the effectiveness of interventions.

Question components are illustrated in Box 1.

(1) What are the effects of individual characteristics, characteristics of the testing process, and healthcare organisational factors on the psychological associations of cancer testing?

(2) What interventions are effective at reducing the adverse psychological associations of cancer testing?

Box 1Inclusion criteria for questions 1 and 2.

**
*Question 1*
**


***Sample:** Adults.*

***Phenomenon of interest:** Testing for cancer (any type).*

***Design of studies:** Cross-sectional, longitudinal (cohort), and mixed-method studies.*

***Evaluation:** Any measure of psychological burden such as worry, anxiety, fear, distress, depression, and uncertainty measured via tools including but not restricted to STAI, HRQoL, SF-12, SF-36, and HADS.*

***Research type:** Quantitative (cross-sectional, case control, and cohort) and mixed-methods, primary studies, or systematic reviews.*


**
*Question 2*
**


***Population:** Adults undergoing diagnostic testing for any type of cancer.*

***Intervention:** Any intervention(s) to improve psychological burden such as worry, anxiety, fear, distress, depression, and uncertainty measured via tools associated with testing for cancer.*

***Control:** No intervention(s) or alternative intervention(s), including standard care.*

***Outcome:** Any measure of psychological burden such as worry, anxiety, fear, distress, depression, and uncertainty measured via tools incuding but not restricted to STAI, HRQoL, SF-12, SF-36, and HADS.*

***Study Design:** Systematic reviews of RCTS or RCTs.*


### 2.2. Search Strategy

Electronic bibliographic databases were searched using a combination of MESH and free-text terms combined using Boolean operators (and/or). OVID MEDLINE, PubMed, Embase, the Cochrane Library, and ClinicalTrials.gov were searched for published articles, and the British Library, Library Hub Discover, Opengrey.eu, the Grey Guide, gov.uk (news and communications), and the National Grey Literature Collection for unpublished literature. Electronic database searches were supplemented with searches of reference lists of included systematic reviews and primary studies. All articles from inception to December 2020 were included. Only articles published in English were included. The search strategy is available as an appendix (Appendix A). This systematic review was prospectively registered on PROSPERO (Registration number CRD42022321906).

### 2.3. Study Selection

Titles, abstracts, and full texts of potentially relevant titles and abstracts were screened by one reviewer against predefined inclusion criteria (Box 1), and reasons for exclusion of studies were documented using a PRISMA flow diagram (Figure 1).

For question 1, our plan was to include the results of systematic reviews relevant to our research question and in addition, if the included reviews were of sufficient quality and relevance, to supplement these with studies published since the review literature search completion dates. However, the systematic reviews we identified as relevant to our research questions synthesised a mix of quantitative and qualitative results, and as we were only interested in quantitative research, we were unable to use their synthesised results. We therefore decided to incorporate the quantitative evidence in the reviews by considering the results of primary quantitative studies included in the reviews (if they were not identified from our own searches).

### 2.4. Data Extraction

A single data extraction form was designed for questions 1 and 2. Data extracted included title, first author, year of publication, study design, aim of study, number of studies/participants, population characteristics, cancer type under investigation, test, intervention (where appropriate), comparator (where appropriate), and results.

### 2.5. Quality Assessment

For quality assessment of systematic reviews, five criteria drawn from the Joanna Briggs Institute (JBI) Checklist for Systematic Reviews and Research Syntheses were assessed, namely, the inclusion of a clear, focused question, clear question formulation, comprehensive search strategy, quality assessment of studies, and data extraction by two independent reviewers [7].

For cross-sectional studies, a modified JBI Checklist for Analytical Cross-Sectional Studies [8] was used: the domain (‘was the exposure measured in a valid and reliable way’) was not considered relevant to this review question and was omitted. For RCTs, the Cochrane Risk of Bias 2 (RoB 2) tool [9] was employed.

We did not identify any cohort or mixed-method studies to include in this review. Quality assessment of primary studies was undertaken in duplicate by FK and CD.

### 2.6. Data Synthesis

Data synthesis was narrative and supported by tables to map similarities and differences in population, cancer, test type, intervention (where applicable), and outcomes for each of questions 1 and 2. Recognising that psychological associations are likely to be different in screening compared to diagnostic applications of testing, these different testing applications were considered separately for the purposes of synthesis.

On the basis of research identified as part of our scoping review of the predictors of anxiety associated with diagnostic and screening tests [10,11,12], we used three themes as the framework for the synthesis of this review: individual characteristics, testing-related factors, and organisational factors.

## 3. Results

### 3.1. Volume of Studies


**Question 1: Psychological associations of testing.**


A total of 26 studies, including 10 systematic reviews (SRs), 15 cross-sectional studies, and 1 randomised controlled trial (RCT) were identified. Of the 10 SRs, testing was undertaken for screening (7 studies), diagnosis (2 studies), or both (1 study). Nine primary studies were concerned with screening whilst seven were concerned with diagnostic testing. 


**Question 2: Effectiveness of interventions to mitigate adverse psychological associations of testing.**


Thirteen randomised controlled trials (RCTs) were included. Interventions were undertaken for screening (five studies) and diagnosis (eight studies).

### 3.2. Characteristics of Included Studies (Appendix A)


**Question 1: Psychological associations of testing.**


SRs from the following countries were included: U.S.A. (five), Finland (one), Australia (one), Ireland (one), The Netherlands (one), and Canada (one). The total number of studies included in each SR (qualitative and quantitative) ranged from 7 to 59, and the number of subjects ranged from 872 to 199,906. Most SRs focused on single cancers, namely, breast (three), cervical (two), colorectal (two), pancreatic (one), and lung (one), whilst one included various cancers.

Quantitative primary studies from Europe (eight), the U.S.A. (three), Taiwan (one), Australia (one), Oman (one), Canada (one), and Lebanon (one) were included. The number of subjects ranged from 31 to 3671. Studies were concerned with testing for cancer of the cervix (seven), breast (six), prostate (one), and ovary (two). Studies included a variety of tests, and different elements of the testing process including mammography (four), colposcopy (four), notification of abnormal cervical smear results (three), biopsy (two), transvaginal ultrasound scan (two), and HPV testing (one). The severity of psychological outcomes was measured at different time points including before testing (four), on the day of testing (seven), and immediately after testing or after receiving the test results (five). Psychological associations were assessed through various validated tools such as PCQ, STAI, COS-BC, SF-12, HADS, and MBSS, as well as author-designed questionnaires, or a combination of these. 


**Question 2: Effectiveness of interventions to mitigate adverse psychological associations of testing.**


RCTs from the U.S.A. (five), Europe (five), Australia (one), Cameroon (one), and Thailand (one) were included. The number of participants ranged from 16 to 838. Interventions were associated with mammography for breast cancer (two studies), diagnostic or interventional colposcopy for cervical cancer (six studies), colonoscopy and flexible sigmoidoscopy for bowel cancer (one study), faecal occult blood test for bowel cancer (one study), a combination of different tests (one study), and biopsies (two studies).

### 3.3. Quality Assessment


**Question 1: Psychological associations of testing (Table 1 and Table 2**
**).**


Aside from three reviews [13,14,15] where it was unclear whether the data extraction and quality assessment had been conducted in duplicate, SRs were considered at low risk of bias on the remaining four quality criteria (Table 1).

In the 15 included cross-sectional studies, all clearly defined the inclusion criteria, study subject, and settings, described how the psychological outcomes were measured, and processed the results using appropriate statistical analysis. Of these studies, 12/15 (80%) utilised validated measurement tools, while 3/15 (20%) measured outcomes using open-ended questions concerning the patients’ emotions in addition to quantitative measurements. Only 1/15 (7%) of studies reported on confounders (Table 2).

**Table 1 cancers-15-03335-t001:** Quality assessment for systematic reviews for question 1 (adapted from the Joanna Briggs Institute (JBI) Checklist for Systematic Reviews and Research Syntheses).

	Clear, Focused Question	Comprehensive Search Strategy ^†^	- **Explicit Criteria for Paper Inclusion** - **Two Independent Reviewers**	- **Explicit Criteria for Quality Assessment** - **Explicit Criteria for Data Extraction**	Validated Methods for Data Analysis	Description of Methods Included and Reproducible
Cazacu et al., 2019 [16]	Yes	Yes	Yes	Yes	Yes	Yes
Chad-Friedman et al., 2017 [5]	Yes	Yes	Yes	Yes	Yes	Yes
Metsälä et al., 2011 [13]	Yes	Yes	Yes	Yes to selection NA to two reviewers	Yes	Yes
Montgomery et al., 2010 [14]	Yes	Yes	Yes to selection NA to two reviewers	Yes to selection NA to two reviewers	Yes	Yes
Nagendiram et al., 2018 [15]	Yes	Yes	Yes	Yes to selection NA to two reviewers	Yes	Yes
Nelson et al., 2016 [17]	Yes	Yes	Yes	Yes	Yes	Yes
O’Connor et al., 2016 [18]	Yes	Yes	Yes	Yes	Yes	Yes
Van der Veld et al., 2017 [19]	Yes	Yes	Yes	Yes	Yes	Yes
Wu et al., 2016 [20]	Yes	Yes	Yes	No	Yes	Yes
Yang et al., 2018 [21]	Yes	Yes	Yes	Yes	Yes	Yes

^†^ The authors provided evidence of a logical and reproducible search strategy which identified the PICO components of the question. More than one citation database including grey literature was searched.

**Table 2 cancers-15-03335-t002:** Quality assessment for cross-sectional studies for question 2 (using the JBI Checklist for Analytical Cross-Sectional Studies).

	Inclusion Criteria Clearly Defined	Study Subjects and Setting Were Clearly Defined	Exposure Measured in Valid and Reliable Way	Measurement of Condition (Were Patients Selected According to Strict Definitions)	Confounders Identified	Strategies to Deal with Confounders Identified	Outcomes Measured in Valid and Reliable Way	Appropriate Statistical Analysis
Al-Alawi et al., 2019 [22]	Yes	Yes	NA	Yes	No	No	Yes	Yes
April-Sanders et al., 2018 [23]	Yes	Yes	NA	Yes	No	Yes	Yes	Yes
Bekkers et al., 2002 [24]	Yes	Yes	NA	Yes	No	Yes	Unclear	Yes
Bolejko et al., 2015 [25]	Yes	Yes	NA	Yes	No	Yes	Yes	Yes
Drolet et al., 2011 [26]	Yes	Yes	NA	Yes	No	Yes	Unclear	Yes
El Hachem et al., 2019 [27]	Yes	Yes	NA	Yes	No	Yes	Unclear	Yes
French et al., 2006 [28]	Yes	Yes	NA	Yes	Yes	Yes	Yes	Yes
Gray et al., 2006 [29]	Yes	Yes	NA	Yes	No	Yes	Yes	Yes
Kola et al., 2012 [30]	Yes	Yes	NA	Yes	No	Yes	Yes	Yes
Liao et al., 2008 [31]	Yes	Yes	NA	Yes	No	No	Yes	No
Maissi et al., 2004 [32]	Yes	Yes	NA	Yes	No	Yes	Yes	Yes
Medd et al., 2005 [33]	Yes	Yes	NA	Yes	No	Yes	Yes	Yes
O’Connor et al., 2016 [34]	Yes	Yes	NA	Yes	No	Yes	Yes	Yes
Wiggins et al., 2017 [35]	Yes	Yes	NA	Yes	No	Yes	Yes	Yes
Wiggins et al., 2019 [36]	Yes	Yes	NA	Yes	No	Yes	Yes	Yes


**Question 2: Effectiveness of interventions to mitigate adverse psychological associations of testing (Figure 2).**


A total of 46% (6/13) of the RCTs were at ‘high’ risk of bias, 31% (4/13) at ‘some concerns’, and 23% (3/13) at ‘low’ risk of bias. For those studies regarded as being at high risk of bias, this was attributed to two domains: the randomisation process and the outcome measurement. The studies were graded as ‘low’ or ‘some concerns’ for the risk of bias across the remaining domains because of one or more deviations from the intended intervention, missing outcome data, and selective reporting of results.

### 3.4. Synthesis of Results


**Question 1: Psychological associations of testing.**


For synthesis, predictive factors were divided into three categories derived from themes identified in the literature. Included studies investigated the association of the following variables in each of the three predefined categories: psychosocial (age, ethnicity, educational status, personal or family history of cancer, employment status, perceived risk of cancer, presence of partner and children, social support, knowledge of cancer, smoking history, and intrinsic trait anxiety), testing-related factors (cancer site, previous abnormal result or severity of index result, procedure-related anxiety, and previous adverse experience of testing), and organisational factors (satisfaction with information received, waiting times, and communication of results). Psychological outcomes including anxiety, depression, distress, or worry were measured using validated measurement tools such as STAI, HADS, Impact of Events Scale (IES), and General Health Questionnaire (GHQ). Individual reported levels of uncertainty, coping style, and expectations were assessed using various tools including the Psychological Consequences Questionnaire (PCQ), Consequences of Screening–Breast Cancer (COS-BC), Positive and Negative Affect Schedule (PANAS), Multi-Dimensional Health Locus of Control Scale (MHLCS), Miller Behavioral Style Scale (MBSS), and Mishel Uncertainty in Illness Scale (MUIS) questionnaires. Fear associated with the testing procedure was measured, e.g., pain was measured using visual analogue scales (VAS) or the Fear of Pain Questionnaire-III (FPQ-III). Finally, the consequences of testing on patients’ quality of life were examined using the EuroQol or Short Form-12 tools.


**I. Individual (psychosocial) characteristics (Appendix A)**



**1. Age**



**
*Screening*
**


Two SRs (including seven studies [16] and seven studies [19] each) and three cross-sectional studies found a negative association between age and psychosocial morbidity in screening for breast [22,23], pancreatic [16], cervical [29], and colorectal [19] cancers.

Four cross-sectional studies found no statistically significant association between age and psychological morbidity with cancer screening for breast [25,27], cervical [26], or ovarian cancers [35].

Two SRs (with 2/15 studies including age as a variable [13] and 5/58 studies including age as a variable [21] in each study) reported conflicting results towards the associations of age on psychological morbidity in breast cancer [13] and colorectal cancer [21] screening.


**
*Diagnosis*
**


Three cross-sectional studies [24,30,34] and one RCT [46] in colposcopy for cervical cancer testing showed no correlation between age and levels of anxiety. One cross-sectional study in breast cancer [31] concluded that age was not a significant predictor for short- or long-term anxiety during the diagnostic phase for women with suspected breast cancer.

One SR including 30 studies [14] reported the role of age as inconclusive. 


**2. Ethnicity**



**
*Screening*
**


One SR [15] on cervical cancer (13 studies), one SR [5] on a combination of cancer types (22 studies), and one cross-sectional study on breast cancer [25] demonstrated that non-white or non-native women were at high risk of psychological distress compared to native or Caucasian women.

In one cross-sectional study on cervical cancer testing [29], ethnicity was not shown to be associated with anxiety following an abnormal cervical smear result.


**
*Diagnostic*
**


One cross-sectional study [34] demonstrated that non-Irish participants were at greater risk of anxiety from cervical cancer testing. 


**3. Education status**



**
*Screening*
**


Three SRs (including 15 studies [13], 13 studies [20], and 58 studies [21] each) and three cross-sectional studies showed a negative association between educational status and anxiety levels in breast [13,25], lung [20], cervical [26], colorectal [21], and ovarian [35] cancer testing. One cross-sectional study [22] found no association between literacy levels and the magnitude of anxiety in women who underwent mammograms for breast cancer screening.


**
*Diagnostic*
**


One SR [21] on the associations of endoscopic procedures for CRC screening showed a negative correlation between education levels and levels of anxiety. Three cross-sectional studies did not find an association between educational level and anxiety in testing for cervical [24,30] and breast cancer [31].


**4. Previous experience of cancer**



**
*Screening*
**


One SR [20] and two cross-sectional studies described a positive association between a family history of cancer and anxiety associated with testing across lung [20], breast [22], and ovarian [35] cancers. 

A single study concerned with the association of previous cancer testing included in the SR by Metsälä [13] did not find an association between a family history of breast cancer and anxiety levels.


**
*Diagnostic*
**


Three studies concerned with the association of previous cancer testing in an SR by Montgomery [14] demonstrated a statistically significant positive correlation between a history of breast cancer and reported levels of distress and anxiety among women awaiting a breast biopsy or curative surgery.


**5. Employment**



**
*Screening*
**


Two cross-sectional studies demonstrated a negative association between employment status and anxiety levels during breast [22] and cervical [29] cancer screening. 


**6. Perceived risk of cancer**



**
*Screening*
**


One SR [16] in pancreatic cancer and three cross-sectional studies in breast cancer [23,25,32] showed a positive association between a perceived risk of cancer and testing. 


**
*Diagnostic*
**


One cross-sectional study [31] demonstrated that a self-perceived probability of breast cancer was associated with statistically higher levels of anxiety before the biopsy but not after a diagnosis of breast cancer.


**7. Social support including living with a partner**



**
*Screening*
**


Two cross-sectional studies in breast [25] and cervical [26] cancer screening demonstrated a positive association of social support on improved psychological outcomes. One cross-sectional study did not find an association between social support and anxiety levels in women following a false positive ovarian cancer screening result [35]. 


**
*Diagnostic*
**


Two cross-sectional studies in cervical cancer [24,30] and one in breast cancer [31] demonstrated that having a partner was protective against anxiety with a statistically significantly lower mean state anxiety score. 

Montgomery et al. [14], in their SR (30 studies), did not find an association between marital status and psychological distress.


**8. Having children**



**
*Screening*
**


One cross-sectional study in cervical cancer screening [29] showed that having children was associated with higher levels of anxiety following an abnormal cervical smear test.


**
*Diagnostic*
**


In one cross-sectional study on cervical cancer [30], parous women were at higher risk of colposcopy-associated distress.

One cross-sectional study [24] and one RCT [46] did not find a correlation between having children and its association on anxiety with colposcopy for cervical cancer. 


**9. Own knowledge of cancer**



**
*Screening*
**


Three cross-sectional studies in breast [25] and cervical [26,32] cancers showed that a lack of knowledge about cancer had a positive association with anxiety levels. 


**10. Smoking status**



**
*Screening*
**


One SR [20] in lung cancer (13 studies), one SR [5] across various cancers, and two cross-sectional studies in cervical cancer [26,29] showed a positive association between smoking status and anxiety levels with cancer testing.


**
*Diagnostic*
**


One RCT and one cross-sectional study demonstrated a positive correlation between smoking and colposcopy for cervical cancer [34,46].


**11. Trait or intrinsic anxiety and depression**



**
*Diagnostic*
**


Five studies included in an SR by Montgomery et al. [14] showed that amongst women referred for colposcopy, those with higher baseline depression scores experienced higher levels of anxiety and depression as well as a greater fear of cancer at the two-year follow-up.


**II. Test-related factors**



**1. Previous experience of testing including severity of initial result**



**
*Screening*
**


Two SRs (including 15 studies [13] and 58 studies [21] each) and two cross-sectional studies [26,27] reported a positive association between a previous adverse experience of testing and more severe initial results in breast cancer [13,27], CRC [21], and cervical cancer [26]. One cross-sectional study in cervical cancer [29] did not find an association between the index smear result or number of previous abnormal results and anxiety levels in cervical cancer testing.


**
*Diagnostic*
**


A positive association between a previous negative experience and anxiety levels in cervical cancer testing was demonstrated in one SR [18] (16 studies). Two cross-sectional studies concerned with colposcopy for cervical cancer, however, did not demonstrate an association between previous results and anxiety levels associated with them [24,30]. 


**2. Procedure-related**


Intimate and invasive examinations have been significantly and positively associated with higher fear, worry, embarrassment, and worries about potential sequelae across breast, colorectal, lung and cervical cancers [13,15,19,21,34]. Various procedures such as HPV testing, colonoscopy, flexible sigmoidoscopy, and prostate needle biopsy were considered in these studies. 


**III. Organisational**



**Information about testing**



**
*Screening*
**


One cross-sectional study showed that lower satisfaction levels with the information from healthcare professionals in women with a false-positive screening mammography for breast cancer [25] were significantly and positively associated with a greater sense of dejection, anxiety, and poorer sleep. In another cross-sectional study [26] on cervical cancer, women who received their abnormal smear results in person reported higher levels of anxiety than those informed by letter or telephone.


**
*Diagnostic*
**


A cross-sectional study by Bekkers [24] indicated that longer waiting times were statistically positively associated with anxiety in women attending for colposcopy for cervical cancer testing. The authors also concluded that there was a statistically significant association between satisfaction with the information from the GP or gynaecologist and the mean state anxiety scores in those women. 


**Conclusions for Question 1**


Several variables were identified which could have positive correlations with anxiety levels in both screening and diagnostic testing for cancer, namely, being non-white or non-native, a perceived higher risk of developing a malignancy, lack of social support, a positive smoking history, and low educational attainments. In breast and cervical cancers, a lack of knowledge about cancer or the testing process was associated with higher anxiety levels in screening populations only. On the other hand, having a partner was protective in screening and diagnostic testing for cervical cancer. The effect of age was inconsistent even within the same cancer (i.e., some breast cancer studies showed lower age was associated with high anxiety levels, while others showed no difference) for both screening and diagnostic tests.

With regard to testing-related factors, the absence of a previous abnormal test result or the receipt of a severe initial screening result were associated with worse psychological outcomes in screening but not diagnostic testing for cervical and breast cancers. Intimate or invasive modalities such as biopsy, colposcopy, or flexible sigmoidoscopy were associated with high anxiety levels during screening and diagnostic testing. A previous adverse experience of testing was associated with worse anxiety levels in breast, cervical, and colorectal cancer testing.

Finally, some organisational practices could be associated with higher anxiety levels: women who received their results in person and those who experienced longer waiting times for a colposcopy following abnormal smears reported higher anxiety levels. A lack of information about testing and subsequent lack of satisfaction also contributed to greater psychological morbidity.


**Question 2. What interventions are effective at reducing the adverse psychological effects of cancer testing? (Table 3)**


A total of 13 RCTs concerned with three cancers (breast, cervical, colorectal) were identified. The studies included screening (five studies) and diagnostic testing (eight studies). Interventions were assigned to five categories: use of information aids, music therapy, livestreaming of real-time videos during colposcopy, organisational factors (patient navigators, one-stop clinics), and pharmaceutical and homeopathic therapies. Psychological outcomes including anxiety, distress, depression, and worry were measured using validated tools such as the STAI, HADS, and IES questionnaires, author-designed questionnaires, or a combination of these. These outcomes were assessed at a single time point (at referral, before, during, or after receiving the intervention) or at two or more time points. 


**1. Use of information aids**



**
*Screening*
**


One RCT in breast cancer testing reported on the effectiveness of information aids in the form of DVDs or printed materials. Hersch [45] demonstrated a significant reduction in anxiety levels for women undergoing mammography with a significant reduction in breast cancer worry in the intervention arm. 


**
*Diagnosis*
**


One RCT [49] showed a statistically significant reduction in STAI scores with the use of an education pamphlet for women undergoing colonoscopy, while de Bie [41] did not find a clinically or statistically significant improvement in STAI scores in women attending for colposcopy.


**2. Music therapy**



**
*Diagnosis*
**


Four RCTs were identified associated with cervical (two), breast (one), and colorectal (one) testing.

Chlan [39] demonstrated that music therapy was associated with a significant decrease in STAI scores in those attending for a flexible sigmoidoscopy. 

Three RCTs [38,42,47] did not demonstrate a significant effect of music therapy on anxiety levels in women undergoing cervical biopsies, colposcopy, or mammography, respectively.


**3. Real-time videos during colposcopy**



**
*Diagnosis*
**


The two RCTs [37,46] which assessed the effectiveness of real-time videos in women attending for visualisation of the cervix following an abnormal smear result both failed to show a significant difference in STAI scores between both arms both before and after the procedure. 


**4. Organisational**



**
*Diagnosis*
**


One RCT reporting on interventions during breast cancer testing [44] concluded that the presence of a patient navigator and an immediate communication of results may be helpful in lowering patient anxiety. One RCT [43] showed that one-stop clinics whereby women attending for breast cancer testing underwent investigations and received their results on the same day compared to women seen in the usual pathway was only beneficial in the short term (24 h) but not at follow-up after three weeks or three months. 


**5. Pharmacological and homeopathic therapies**



**
*Diagnosis*
**


One RCT assessing homeopathy in women undergoing breast biopsies [48] noted a significant decline in anxiety with hypnosis and relaxation techniques following this intervention. Cruickshank [40], on the other hand, did not find any significant change in the HADS scores with the self-administration of an inhaled general anaesthetic (isoflurane) in women attending for colposcopy. 


**Conclusions for Question 2**


Most RCTs were conducted in diagnostic populations for breast and cervical cancer. Of the five intervention categories, the use of information aids and organisational modifications such as the introduction of a patient navigator or one-stop clinics appeared to reduce anxiety. Homeopathic and complimentary therapies such as hypnosis may also be helpful. On the other hand, there was minimal evidence to support the use of music therapy or livestreaming of real-time videos during colposcopy. Overall, there is a paucity of evidence to support the majority of the interventions under consideration in this review for any cancer type or testing process.

## 4. Discussion

### 4.1. Summary of Findings

Some individual variables such as a real or perceived lack of knowledge of cancer testing, current or previous smoking history in lung and cervical cancer testing, and higher levels of trait anxiety as well as the invasive or intimate nature of some testing modalities (colonoscopy, flexible sigmoidoscopy, prostate needle biopsy, and colposcopy) have consistently been demonstrated to be associated with higher levels of fear, worry, embarrassment, and anxiety across various cancers (breast, colorectal, prostate, lung, and cervical cancers).

Our review suggested that cultural factors, language, and religious beliefs in women from non-white and immigrant communities may hinder attendance for cervical and breast cancer testing. However, the relevance of ethnicity as a risk factor for higher anxiety remains debatable in view of the variation in cancer types, testing modalities, study designs, and paucity of details with respect to the country of origin, refugee status, or ethnicity for non-native women. Similarly, the role of education remains unclear as the definitions used to report on different levels of education were non-uniform across the included studies. The relevance of age as a risk factor for anxiety was assessed in four SRs and 12 primary studies involving 3444 subjects. Its effect was unclear even within the same cancer. For instance, some BC studies showed that lower age was associated with high anxiety, while others showed no difference. This could be attributed to the heterogeneity in the age thresholds used to triage the subjects into ‘younger’ and ‘older’ categories across the included studies.

For interventions to mitigate the psychological associations of testing, our study appeared to confirm the effectiveness of informational aids which already constitute an integral part of patient care in a clinical setting across the U.K. Other, relatively novel organisational factors such as patient navigators or one-stop clinics seemed to play a role in mitigating anxiety levels. The evidence to support music therapy or real-time videos, which are an integral aspect of the majority of colposcopy clinics across the U.K., was less robust.

### 4.2. Strengths and Limitations of Review Methods

To our knowledge, this is the first review which addresses the harms of cancer testing and evaluates the effectiveness of interventions to mitigate their associations. We conducted a comprehensive literature search and undertook quality assessment in duplicate. We acknowledge a limitation of our review methods is that screening, inclusion, and data extraction were conducted by a single author. 

All the included SRs concerned with the psychological association of testing were narrative, and none of these offered a quantitative assessment of the results; it was therefore not appropriate to perform a meta-analysis. In addition, heterogeneity of cancer type, test, and interventions precluded meta-analysis. Heterogeneity of results even within similar populations and testing modalities may be explained by differences in outcome measurement. The outcome ‘psychological distress’ was used broadly and was often used interchangeably with anxiety, depression, stress, and distress across studies. The lack of a universal definition has resulted in the use of a broad spectrum of validated measurement tools including author-designed questionnaires. Finally, study quality was variable and a main limitation across the included primary studies in question was the non-identification of confounders, which undermines the validity of analyses. 

The divergence in results across studies could thus be attributed to the lack of consistency pertaining to the heterogeneity across cancer types, measurement tools, definition of psychological distress, and time points at which psychological distress was assessed across studies.

With regard to question 2, most of the randomised controlled trials were conducted in patients attending for diagnostic rather than screening tests and therefore address the mismatch in the existing body of the literature, which is more well-researched in the context of screening tests. However, the interventions to mitigate anxiety were primarily conducted for cervical, breast, and colorectal cancers, that is, cancers with an established screening programme in the U.K. The latter were overrepresented compared to cancers with a lower incidence but which are often lethal, such as ovarian, pancreatic, and lung cancers. Further research into methods to address individuals at risk of, or with symptoms suggestive of, these more lethal cancers would undoubtedly be more helpful to increase attendance for investigations, improve uptake of testing, and may perhaps improve survival through earlier diagnosis. 

Finally, the most recent primary studies identified for both questions 1 and 2 were conducted in 2019. We acknowledge that the evidence is likely to have progressed since the completion of this review, thereby impacting on its currency. 

### 4.3. Clinical Implications

The psychological benefit of cancer testing, namely, the reassurance afforded by an estimation of the patient’s risk value for cancer following testing, has been described in the literature [50]. In addition to this, the mortality benefits of population-based screening for breast, lung, colorectal, and cervical cancers have been previously demonstrated [51,52,53,54]. Although screening for ovarian cancer in asymptomatic average-risk women does not confer any survival benefit, further studies are underway to explore whether diagnostic testing of women who present to their doctors with suspicious symptoms could be associated with a survival benefit. To this end, there is an urgent need to investigate the psychological associations of cancer testing. Existing research is focused on screening for a select number of cancers such as cervical, breast, and colorectal. However, certain fatal and deadly cancers such as ovarian and pancreatic cancers are underrepresented in the existing research portfolio. Evidence from our study suggests that the roles of some individual characteristics (age, ethnicity, educational attainments, employment status, and marital status) warrant further research to understand whether they are modifiers of the psychological associations of cancer testing. Assessment of the applicability of findings is further limited in view of the different screening and testing pathways employed in different countries. In terms of research methods, this area of research poses difficulties. For instance, it is difficult to blind participants and their assessors to interventions to mitigate anxiety in these testing contexts, especially if these involve non-concealable methods such as music therapy or real-time videos. Bias introduced by evaluation of outcome questionnaires may possibly be addressed to improve blinding of outcome measurement.

The result of our literature review suggests that some individual variables such as a real or perceived lack of knowledge of cancer testing, risk behaviours, higher levels of trait anxiety, and the invasive or intimate nature of some testing modalities have consistently been demonstrated to be associated with higher levels of fear, worry, embarrassment, and anxiety. These variables associated with testing encounters could be targeted for any interventions to mitigate the adverse psychological outcomes associated with cancer testing.

Our research demonstrates that modifiable (organisational) factors such as one-stop clinics and patient navigators for intervention evaluation may be beneficial in patients attending for cancer testing. With regard to interventions to mitigate anxiety, shifting towards one-stop clinics represents a potential route to expedite diagnosis and may thereby be helpful to reduce the anxiety associated with prolonged waiting times. Continued use of information aids to educate patients about the cancer under review and the nature of and potential outcomes from associated investigations should be encouraged.

## 5. Conclusions

In summary, this literature search has identified some potential variables which may be associated with psychological morbidity in both screening and diagnostic cancer testing applications. Targeting certain patient groups and testing situations may offer a means to mitigate anxiety. Certain interventions may be helpful to mitigate the psychological morbidity associated with testing. A limited body of research suggests that one-stop clinics and patient navigators may be beneficial in patients attending for cancer testing. The contribution of some factors to anxiety in cancer testing and their specificity of effect are inconclusive and warrant further research in homogenous populations and testing contexts.

## Figures and Tables

**Figure 1 cancers-15-03335-f001:**
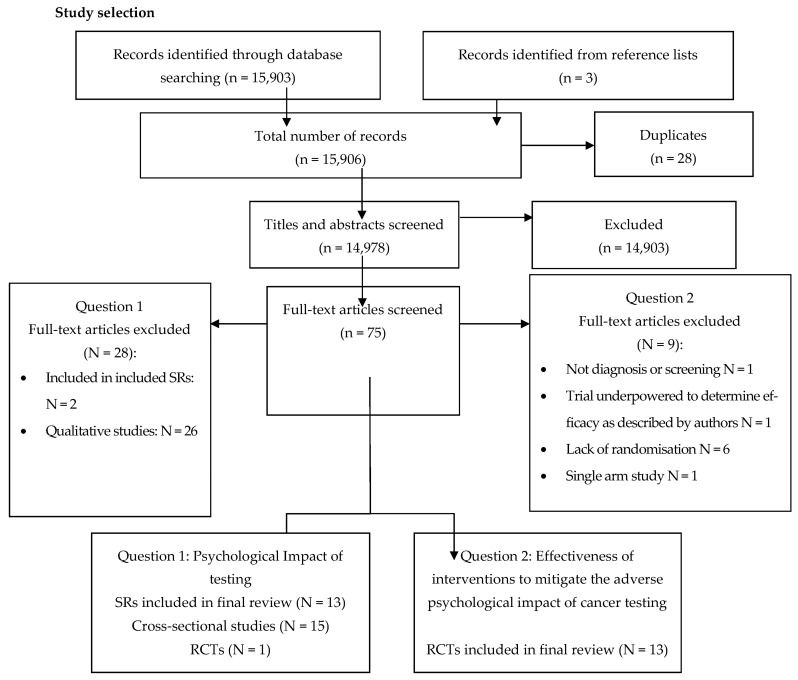
PRISMA flow diagram.

**Figure 2 cancers-15-03335-f002:**
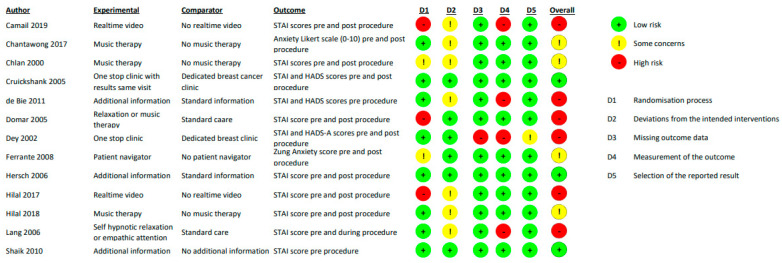
Quality assessment for randomised controlled trials for question 3 (using the Risk of Bias RoB2 tool). Camail 2019 [37]; Chantawong 2017 [38]; Chlan 2000 [39]; Cruickshank 2005 [40]; de Bie 2011 [41]; Domar 2005 [42]; Dey 2002 [43]; Ferrante 2008 [44]; Hersch 2006 [45]; Hilal 2017 [46]; Hilal 2018 [47]; Lang 2006 [48]; Shaik 2010 [49].

**Table 3 cancers-15-03335-t003:** Results of RCTS: interventions to reduce psychological morbidity associated with cancer testing for question 2.

Author, Year	Cancer Test	Intervention (I)/ Control (C)	Measurement Tools and Assessment Timepoints	Effect Size	
Camail et al., 2019 [37]	Cervix Visual inspection of cervix	I: Realtime video during (N = 60) C: No realtime video (N = 58)	Measurement tools: STAI Timepoints: Pre and post procedure	mean (S.D) Intervention Control	Before: 36.4 (11.8) After: 28.5 (12.0) ∆ = −7.9 (14.3.), *p*-value < 0.05 Before: 33.6 (10.9) After: 29.3 (11.2) ∆ = −4.2 (9.0), *p*-value < 0.05 ∆I and C (before and after), *p*-value = 0.10
Chantawong et al., 2017 [38]	Cervix Loop *electrosurgical excision* procedure (LEEP)	I: Music (N = 74) C: No music (N = 76)	Measurement tools: VAS Timepoints: Pre and post LEEP	∆ mean (S.D) I. Pre-biopsy II. Post-biopsy	I: 3.7 (2.6) C: 4.1 (3.0), *p*-value 0.38 I: 4.0 (2.9) C: 4.7 (3.2), *p*-value 0.16
Chlan et al., 2000 [39]	Colon Flexible sigmoidoscopy	I: Music during flexible sigmoidoscopy (N = 30) C: No music during flexible sigmoidoscopy (N = 34)	Measurement tools: STAI Timepoints: Pre and immediately post procedure	∆ mean (SD) Baseline state anxiety Procedure state anxiety	I: 36.9 (12.5) C: 40.2 (11.9), *p*-value 0.28 I: 34.5 (10.0) C: 41.8 (13.5), *p*-value 0.02
Cruickshank et al., 2005 [40]	Cervix colposcopy	I: Self-administration of isoflurane and desflurane (N = 198) C: Self-administration of placebo (N = 198)	Measurement tools: HADS (anxiety subscale) Timepoints: - At baseline prior to treatment - Immediately posttreatment - 6 months after treatment but prior to receiving a colposcopy follow-up appointment	∆ mean (S.D) Baseline Immediately posttreatment 6-month follow-up	I: 8.37 (4.15) C: 7.77 (3.97), *p*-value ‘ns’ * I: 7.30 (4.11) C: 7.29 (4.06), *p*-value ‘ns’ I: 6.49 (4.19) C: 6.49 (4.27), *p*-value ‘ns’
De Bie et al., 2011 [41]	Cervix Colposcopy	I: Individually targeted information (N = 75) C: Standard care (N = 74)	Measurement tools: - HADS - STAI Timepoints: Prior to colposcopy	∆ median (IQR) STAI HADS anxiety	I: 33.0 (27.0–41.0) C: 33.0 (27.0–41.3), *p*-value 0.96 I: 5.0 (3.0–9.0) C: 6.0 (4.0–10.0), *p*-value 0.26
Dey et al., 2002 [43]	Breast Mammography, USS, aspiration cytology, same-day results and management plan.	I: One-stop clinics (N = 267) C: Dedicated breast clinics with women asked to return the following week to discuss the results (N = 211)	Measurement tools: - HADS (anxiety subscale) - STAI Timepoints: - Baseline (immediately before assessment) - STAI (24 hours after first visit) - HADS (three weeks and three months after diagnosis)	∆ mean (S.D) STAI Baseline 24 hours HADS Baseline Three weeks Baseline Three months	I: 48.1 (13.9) C: 47.2 (14.9) I: 34.5 (14.6) C: 39.8 (15.8), *p*-value < 0.0001 I: 8.9 (4.4) C: 8.8 (4.9) I: 7.3 (4.7) C: 7.4 (4.3), *p*-value 0.55 I: 8.9 (4.4) C: 9.0 (5.0) I: 7.0 (4.6) C: 7.5 (4.7), *p*-value 0.22
Domar et al., 2005 [42]	Breast Mammography	I1: Listening to relaxation tape during screening mammography (N = 50) I2: Listening to music tap during screening mammography (N = 47) C: No tape (N = 46)	- STAI - Likert anxiety score Timepoints: At recruitment and immediately after mammography.	∆ mean (SD) STAI Baseline After mammography Likert anxiety score After mammography	I1: 34.8 (9.7) I2: 33.6 (8.9) C: 33.2 (14.5), *p*-value 0.18 I1: 30.4 (9.3) I2: 30.9 (10.0) C: 33.2 (13.3), *p*-value 0.78 I1: 2.6 (1.9) I2: 3.2 (2.3) C: 2.8 (2.4), *p*-value 0.43
Ferrante et al., 2008 [44]	Breast Mammography	I: Patient navigator (N = 55) C: No patient navigator (N = 50)	Measurement tools: Zung Anxiety Self-Assessment Scale Timepoints: At enrolment and 1 month after final resolution (benign diagnosis or for cancer patients, after initiation of cancer treatment).	∆ mean (SD) Baseline Follow-up ∆anxiety with time	I: 38.7 (13.0) C: 36.6 (9.3), *p*-value 0.346 I: 30.2 (7.6) C: 42.8 (13.3), *p*-value < 0.001 I: −8.0 (10.6) C: 5.8 (14.0), *p*-value < 0.001
Hersch et al., 2006 [45]	Breast Mammography	I: Decision aid including evidence-based information about important outcomes of breast screening compared with no screening (N = 419) C: Control decision aid which omitted all overdetection-related content but was otherwise identical (N = 419)	Measurement tools: - STAI - Structured questionnaire Timepoints: - Baseline interview 1-4 weeks after recruitment - Telephone interview 1-4 weeks after being sent decision aid	∆ mean (SD) STAI Questionnaire Worry about breast cancer (%) - Not worried at all - A bit worried - Quite worried or very worried	I: 29·7 C: 29·6 ∆ (95% CI): 0·1 (−1.4 to 1.6), *p*-value 0·93 I: 42, C:32, ∆ (95% CI): 9·4 I: 51, C:55, ∆ (95% CI): −3.9 I: 7, C:13, ∆ (95% CI): -5.5 *p*-value 0.0026
Hilal et al., 2017 [46]	Cervix Colposcopy	I: Realtime video during colposcopy (N = 111) C: No realtime video during colposcopy (N = 105)	Measurement tools: - STAI - VAS Timepoints: - STAI: scores measured before (Appendix A) and after (Appendix A) colposcopy - VAS: anxiety during colposcopy	∆ median (IQR) STAI Before colposcopy (S1) After colposcopy (Appendix A) ∆S (Appendix A) VAS	I: 51 (42–62) C: 50 (41–61), *p*-value 0.73 I: 39 (33–50) C: 40 (33–48.5), *p*-value 0.80 I: −10.3 ± 11.3 C: −10.3 ± 11.0, *p*-value 0.50 I: 3 (1–5) C: 3 (1–5), *p*-value 0.61
Hilal et al., 2018 [47]	Cervix Colposcopy	I: Music during colposcopy (N = 103) C: No music during colposcopy (N = 102)	Measurement tools: - STAI - VAS Timepoints: STAI: scores measured before (Appendix A) and after (Appendix A) colposcopy VAS: anxiety during colposcopy	∆ median (IQR) STAI Before colposcopy (S1) After colposcopy (Appendix A) ∆S (Appendix A) VAS	I: 48 (42–59) C: 50 (41–59.25), *p*-value 0.91 I: 40 (33–47) C: 439 (32.75–46), *p*-value 0.92 I: −9.4 ± 10.8 C: −9.0 ± 10.6, *p*-value 0.40 I: 2 (1–5) C: 2 (1–6), *p*-value 0.28
Lang et al., 2006 [48]	Breast Large core needle biosy	I1: Self-hypnotic relaxation (N = 78) I2: Empathic Attention (N = 82) C: Standard of care (N = 76)	Measurement tools: - Anxiety on VAS of 0–10 (t = 0 to t = 110 minutes) - STAI Timepoints: - Baseline anxiety - Self-rating anxiety every 10minutes in procedure room	Significant increase in anxiety in control No change in empathy group Significant decrease in hypnosis group	logit slope = 0.18, *p* < 0.001 logit slope = −0.04, *p* = 0.45 logit slope = −0.27, *p* < 0.001
Shaik et al., 2010 [49]	Colorectal Colonoscopy	I: Information aid including American Gastroenterological Association colonoscopy educational pamphlet along with prep instructions (N = 51) C: Standard preparationinstructions only (N = 55)	Measurement tools: STAI Timepoints: Immediately before colonoscopy	∆ mean STAI Medication usage Midazolam (reduces anxiety) Meperidine (reduces pain)	I: 40.54 C: 45.18, *p*-value 0.0146 I: 2.35 C: 2.9, *p*-value 0.0444 I: 73.03 C: 76.81, *p*-value 0.374

Abbreviations: measurement tools include CRI—Coping Resources Inventory; COS-LC—psychosocial consequences of lung cancer screening; HADS—Hospital Anxiety and Depression Scale; IES—Impact of Event Scale; Likert anxiety score; SF-12—12-Item Short Form Health Survey; STAI—State–Trait Anxiety Inventory; VAS—*Visual Analogue Scale*; and Zung Anxiety Self-Assessment Scale. Others include LEEP—loop *electrosurgical excision* procedure; USS—ultrasonography; and *‘ns’, which refers to when authors did not report a *p*-value.

## Data Availability

Data sharing not applicable.

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
