# Peer review of "Evaluating the Harms of Cancer Testing—A Systematic Review of the Adverse Psychological Correlates of Testing for Cancer and the Effectiveness of Interventions to Mitigate These"

_cancers, 2023, doi:10.3390/cancers15133335_

Round 1

Reviewer 1 Report

See attached review

A careful editing to improve clarity would be helpful. 

Reviewer 2 Report

I congratulate the authors of this topic. Anxiety and stress in oncological patients are problems that cause limitations in undergoing prophylactic examinations, including screening tests, and also reduce the quality of life of patients during diagnostics. The presented work may guide and help to plan interventions useful in alleviating the negative psychological effects at the stage of performing diagnostic tests for cancer.

Author Response

We would like to take this opportunity to thank the reviewer for taking the necessary time and effort to review our manuscript. We sincerely appreciate all your valuable comments and suggestions without which it would be impossible to maintain the high standards of peer-reviewed journals.

Reviewer 3 Report

I have recently received your manuscript titled Evaluating the harms of cancer testing - a systematic review of the adverse psychological impact of testing for cancer and the effectiveness of interventions to mitigate these for review. After a thorough evaluation, I am pleased to inform you that your work is well written, and I do not have any concerns or additional comments for improvement. I am recommending that your paper be accepted for publication in its current form.

Your study addresses an important topic in the fields of psychology and oncology, and the results and discussion presented in the manuscript contribute significantly to the current understanding of the subject. The review is well-structured, clear, and concise, and it effectively conveys the research findings and their implications.

I am confident that your manuscript will be of great interest to the journal's readership and will contribute to the advancement of knowledge in this area.

Once again, congratulations on your outstanding work, and I look forward to seeing your manuscript published in the journal.

PS one minor remark: the numbers in the Reference list are duplicated. Please check

Author Response

(The authors gave the same response as above.)

Round 2
